# Data-Driven Fouling Detection in Refinery Preheat Train Heat Exchangers Using Neural Networks and Gradient Boosting

**DOI:** 10.3390/s25164936

**Published:** 2025-08-09

**Authors:** Željka Ujević Andrijić, Nikola Rimac

**Affiliations:** Department of Measurements and Process Control, Faculty of Chemical Engineering and Technology, University of Zagreb, Savska c. 16/5A, 10000 Zagreb, Croatia; zujevic@fkit.unizg.hr

**Keywords:** fouling detection, machine learning, preventive maintenance, preheat train, heat efficiency CDU, heat exchanger

## Abstract

Fouling detection in refinery crude distillation unit (CDU) preheat trains is essential for maintaining energy efficiency and operational reliability. This study presents a virtual sensing approach for fouling monitoring using data-driven and semi-empirical models. Specifically, Long Short-Term Memory (LSTM) neural networks, Extreme Gradient Boosting (XGB), and the ɛ-NTU method (effectiveness—Number of Transfer Units) were compared for predicting heat exchanger outlet temperatures, which serve as indicators of fouling. Models were trained on clean operational data to estimate baseline performance. A growing discrepancy between predicted and actual outlet temperatures over time indicated heat transfer degradation. Fouling resistance was calculated from the difference between predicted and actual heat transfer coefficients, enabling effectiveness loss assessment. The LSTM model showed high accuracy in capturing dynamic operational trends, while XGB provided a lightweight alternative with limited extrapolation capability under unfamiliar conditions. Both models outperformed the ɛ-NTU approach in fouling detection sensitivity. Inefficiencies from a single fouled exchanger were estimated to result in an additional 175 tons of CO_2_ emissions and an economic loss of approximately EUR 12,000 over two months. This study highlights the potential of AI-enabled virtual sensors for real-time fouling monitoring in industrial heat exchangers. Such tools can significantly enhance predictive maintenance strategies, improve energy efficiency, and reduce emissions.

## 1. Introduction

It has been a long-lasting challenge for the process industry to reduce energy consumption and emissions in crude oil refineries as one of the biggest energy consumers in the process industry [1]. The Crude distillation unit (CDU) is the central process unit of a refinery and is also one of its most energy intensive operations inside the refinery. In the process of crude distillation, the crude oil enters the distillation column at temperatures between 330 °C and 380 °C which is being reached with the heat exchanger network (HEN) and the furnace [2]. HENs, as key operations for efficiently energy utilization, have been subject to many studies on the optimization and design of networks [3], as well as the processes inside the heat exchangers such as heat transfer efficiency and the formation of fouling depositions [4]. Depositions on the heat transfer surface degrade heat transfer efficiency, which usually brings more primary fuel consumption [5]; therefore, the operators need timely and purposeful information about the behavior in order to react and optimize fuel consumption and carbon emissions. Timely information also gives the operator the ability to schedule cleanings of heat exchangers to minimize operational disruptions [6], thus extending equipment lifetime, avoiding unnecessary shutdowns and expensive repairs. There is a significant amount of research concerning the detection and prediction of fouling, which can be attributed to its serious environmental and financial implications. In the US refining sector alone, heat exchanger fouling in CDUs was projected to cost approximately 1 to 1.2 billion USD in 2019 [7]. Furthermore, a loss of efficiency in heat exchangers leads to negative environmental effects with excessive carbon emissions [8].

In the extensive research on semi-empirical fouling models, Ur Rehman et al. [9] highlighted the differences and advancements between deterministic, threshold, and modern computational fluid dynamics (CFD) simulation models. The authors reviewed methods for determining mechanisms involved in the formation of deposition and mechanism incorporation in semi-empirical models, concluding that great effort has been made by researchers in the field. However, the complexity related to crude oil composition remains a considerable challenge. Additionally, while some of the reviewed CFD simulations made a strong breakthrough, they were primarily based on laboratory-scale experimental and simulated data, lacking validation with field data from heat exchangers in CDUs. Fouling inside HENs is a complex phenomenon due to a lack of understanding of the physical and chemical mechanisms involved [10]. Developing a widely accepted fundamental model has been challenging due to the dynamics of change in operating parameters, system geometry, as well as variations in crude oil used inside real-world refinery plants [11,12]. Another approach was reviewed in the summary by Wilson et al. [13], which provides an overview of various studies on fouling prediction. The authors included an additional group of models—artificial neural network (NN) models for fouling determination. These data-driven models identify correlations between the defined process features and provide sufficient insights into process behavior without entering the underlying mechanism complexity [11]. Although these models have demonstrated notable success in various studies, they often rely on simulated conditions, laboratory-scale experiments, or predefined predict fouling resistance (*R*_f_) values—parameters that are frequently unavailable or impractical to determine in real industrial settings. The methodology presented in this study, on the other hand, is based solely on real operational data collected from refinery heat exchangers under actual process conditions. This avoids the issue of idealized models that cannot capture the operational dynamics and frequent feedstock changes typical of industrial settings. Furthermore, the proposed approach eliminates the need to include *R*_f_ as an output variable in the model, significantly simplifying its implementation in real-time monitoring applications. Instead, the predicted outlet temperature serves as an indicator of exchanger cleanliness, with the gradual deviation from measured values directly reflecting the extent of fouling. This framework allows for the continuous tracking of the heat exchanger performance using only plant-available data and minimal computational effort. These advantages are visible while comparing the proposed method with conventional methods. Conventional methods for monitoring fouling in heat exchangers include tracking changes in heat transfer coefficients, pressure drops, and temperature differences across the unit, as well as using ultrasonic, electrical, or gravimetric techniques. While these approaches are widely used in industry, they have several important limitations. Methods based on heat transfer and temperature require stable operating conditions and provide limited information about the location or nature of fouling. Pressure drop measurements are influenced by the design of the exchanger and process conditions and do not reliably indicate the extent of deposition. Ultrasonic and electrical sensors allow for localized detection but are often expensive, sensitive to temperature changes, and ineffective if fouling occurs outside the monitored area. Gravimetric analysis, although accurate, requires shutdown and disassembly, making it unsuitable for continuous use. These limitations underline the need for more flexible and continuous monitoring strategies, particularly those based on data analysis and machine learning techniques that can account for changing operating conditions [11].

Fouling is defined as the physical and/or chemical deposition of unwanted materials on the heat exchanger surfaces over time [8]. This indicates that the dynamics of deposition is time-dependent, suggesting that valuable information about current fouling can be extracted from the historical data preceding the current state. This research leverages the advantages of LSTM neural networks in terms of capturing time-dependent correlations between input and output data by introducing historical data to the input layer of the model. This methodology is used to develop an effective tool for technologists for the detection of fouling in a highly dynamic environment, using continuously collected field data.

Due to these advantages—including the use of real process data, the exclusion of *R*_f_ from the model outputs, and the retention of operational dynamics—this study represents a meaningful advancement in the development of practical diagnostic tools for industrial heat exchanger monitoring.

Research specifically focused on using LSTM models for fouling detection and prediction in CDU heat exchangers is limited. However, there are some examples of LSTM networks being applied in this context. Madhu et al. [14] used LSTM networks to predict R_f_ in heat exchangers in thermal power plants and refineries. The thickness of the deposition on the heat transfer surface of an aircraft heat exchanger was predicted using LSTMs by Wang et al. [15]. Because of its advantages of remembering the importance of certain historical data, LSTM finds its use in several challenges within chemical process engineering. LSTM’s fault detection capabilities were tested by Kang [16], where LSTMs were utilized to early detect fault operations in complex chemical processes such as the Tennessee Eastman process. Han et al. [17] utilized LSTM in order to analyze the production and energy consumption of two highly complex chemical processes of purified terephthalic acid and ethylene production processes.

In real process operations, the processing time is a variable of great importance. Therefore, a newly developed XGB machine learning model is inspected for this challenge. Introduced by Chen and Guestrin [18] in 2016, XGB is a technique that enhances prediction accuracy and computational efficiency, lowers the risk of overfitting, and requires less training time [19,20]. Its effectiveness has been proven in predicting porosity and permeability in crude oil fields, leading to the better exploration and development of oil and gas resources [21]. Additionally, its superior accuracy has been demonstrated in predicting the adsorption and separation performance of noble gases in metal–organic frameworks [22] and the absorption capabilities of substances harmful to the environment and public health [20]. The application of XGB models in oil and gas chemical process engineering is still an emerging research area, with not many published papers. However, there are several studies showing high accuracy in predicting crude oil prices [23] and its strong performance in forecasting oil and gas production [24].

## 2. Fouling Detection Methodology and Theory

### 2.1. Fouling Detection Methodology

This research proposes a method for fouling detection by predicting clean heat exchanger outlet temperatures for both hot and cold flows using plant data and laboratory tests conducted with different crude oils over the examined period. By relying on real operational data, this calculation inherently accounts for potential imperfections in operating parameters, geometry, and other specific parameters of a real process unit, rather than using values calculated by design equations and heat balances, which typically give idealized results. Models for predicting the temperatures ought to be trained and validated on the data of the process called the initiation phase. The initiation phase is the first of several phases of fouling deposition on the heat transfer surface in which its considered depositions do not affect the heat transfer. It typically lasts for about the first 400 h until the first signs of fouling [25]. The time needed for fouling resistance to return to the zero value before the fouling rate starts to rise in time is called the roughness delay time. The initiation phase is a segment of the full roughness delay time [26]. Figure 1 shows the timespan of four characteristic fouling curves after the roughness delay time: linear (A), falling (B), asymptotic (C), and saw-tooth (D) fouling curves.

As mentioned, data from the initiation phase was used to develop the LSTM, XGB model, and ɛ-NTU method, serving as the training set to predict what the clean heat exchanger outlet temperatures would look like. The models are trained with high accuracy on the dataset, with outlet temperatures of the initiation phase being the targets, and the rest of the selected operational and laboratory data acquired as features. When the new data after the initiation phase was introduced to the model, it would give the predictions of what the model presumes are the clean state outlet temperatures, when in reality, the heat exchanger is already under fouling. Real outlet temperatures are then used for the calculation of the real, dirty overall heat transfer coefficient, and model output clean temperatures are used for the calculation of the clean overall heat transfer coefficient. Reciprocal values of differences between clean and dirty overall heat transfer coefficients are the value of fouling resistance in the examined heat exchanger. Model clean outlet temperatures and real dirty outlet temperatures are also used for the calculations of fouled heat exchanger effectiveness drop. The procedure of the LSTM and XGB model-based fouling and effectiveness calculation method is shown in Figure 2.

The Number of Transfer Units method provides a quick approach for estimating the fouling resistance rate based on the NTU values of a clean operating state of heat exchangers. In this research, the clean state is represented by the above-mentioned initiation phase. The average NTU value (${\bar{NTU}}_{IP}$), calculated from process data during the initiation phase, represents the geometric and hydrodynamic characteristics of that clean phase. However, this approach does not offer validated accuracy based on the input–output patterns of trained data-driven methods; therefore, it is used as an informative comparison tool. This approach is derived from the widely known effectiveness-NTU (ɛ-NTU) method, used and explained in detail by Ujević Andrijić et al. [11].

### 2.2. Heat Exchanger Heat Balances

Typical hot and cold side heat balances are represented by Equations (1) and (2):(1)$$Q_{H}={\dot{m}}_{H}\cdot c_{p,H}\Delta T_{H}$$(2)$$Q_{C}={\dot{m}}_{C}\cdot c_{p,C}\Delta T_{C}$$Q represents heat transfer rate, $\dot{m}$ is mass flow rate, $c_{p}$ is fluid specific heat capacity, and ∆T is a temperature difference in fluids. *H* and *C* subscripts denote hot and cold sides, respectively.

If the heat exchanger is affected by the formation of deposits, the heat transfer decreases, i.e., the efficiency of the heat exchanger decreases, which is given by Equation (3):(3)$$\varepsilon =\frac{Q_{a}}{Q_{max}}$$$Q_{a}$ is the actual delivered amount of heat, and $Q_{max}$ is the maximum possible amount of heat transferred. They are defined by expressions:(4)$$Q_{a}=C_{H}\left(∆T_{H}\right)=C_{C}\left(∆T_{C}\right)$$(5)$$Q_{a}={\dot{m}}_{H}\cdot c_{p,H}\Delta T_{H}={\dot{m}}_{C}\cdot c_{p,C}\Delta T_{C}$$(6)$$Q_{max}=C_{min}\cdot \Delta T_{max}$$$Q_{a}\;$is defined as the product of the mass flow of the fluid, the specific heat capacity, and the temperature difference at the inlet and outlet of the exchanger and is valid for both sides, as seen in Equations (4) and (5). $Q_{max}$ is defined as the product of the minimum value, $C_{min},$ and the maximum temperature difference at a given moment. $C_{min}$ is the product of the mass flow rate and the specific heat capacity of the fluid that achieves a greater temperature difference. From the collected thermodynamic data from the heat exchanger, $C_{min}$ takes the following form:(7)$$C_{min}={\dot{m}}_{H}\cdot c_{p,H}$$$\Delta T_{\max}$ is the largest temperature difference between the inlet temperatures of both fluids, $\Delta T_{\max}=\;(T_{H,\;i}-T_{C,\;i})$, based on relations of inlet and outlet temperatures along the length of the countercurrent heat exchanger.

The obtained relations can further be used to develop an expression for efficiency:(8)$$\varepsilon =\frac{Q_{a}}{Q_{max}}=\frac{{\dot{m}}_{H}\cdot c_{p,H}\Delta T_{H}}{C_{min}\cdot \Delta T_{max}}=\frac{{\dot{m}}_{H}\cdot c_{p,H}(T_{H,i}-T_{H,o})}{{\dot{m}}_{H}\cdot c_{p,H}(T_{H,i}-T_{C,i})}$$
which can then be arranged as(9)$$\varepsilon =\frac{\left(T_{H,i}-T_{H,o}\right)}{\left(T_{H,i}-T_{C,i}\right)}$$The heat transfer can be calculated using(10)$$Q=U\cdot A\cdot \Delta T_{\mathrm{LM}}$$U is the overall heat transfer coefficient, A is the heat transfer area, $∆T_{LM}$ is the logarithmic temperature difference between hot and cold streams given as(11)$$\Delta T_{\mathrm{LM}}=\frac{\left(T_{H,i}-T_{C,o})-(T_{H,o}-T_{C,i}\right)}{\ln((T_{H,i}-T_{C,o})/T_{H,o}-T_{C,i}))}$$The overall heat transfer coefficient is calculated as follows:(12)$$U=\frac{\dot{m}\cdot c_{p}\cdot \Delta T}{A\cdot F\cdot \Delta T_{LM}}$$
where *F* is the correction factor. The fouling resistance is calculated by(13)$$R_{f}=\frac{1}{U_{\mathrm{fouling}}}-\frac{1}{U_{\mathrm{clean}}}$$
where *U*_fouling_ represents the heat transfer coefficient in the presence of fouling, while *U*_clean_ corresponds to the heat transfer coefficient when the heat exchanger is in a clean state. The Number of Transfer Units or NTU is a dimensionless parameter that represents the relationship between the overall heat transfer coefficient, heat transfer area, fluid flow rate, and heat capacity:(14)$$NTU=\frac{U\cdot A}{m{\cdot c}_{p}}$$NTU serves as a valuable efficiency indicator of a heat exchanger, as it integrates geometric and hydrodynamic characteristics [11].

### 2.3. LSTM Architecture

The first LSTM model was presented by Hochreiter and Schmidhuber [27] in 1997 as an upgrade to recurrent neural networks (RNNs) to overcome challenges while learning long-term dependencies. One LSTM unit is structured with a memory cell and input, output, and forget gates defined to selectively regulate irrelevant from relevant information. A typical LSTM unit is shown in Figure 3.

The structure of the LSTM unit is described in the following section [28]: At time *t*, the unit stage consists of the output stage, h(*t*), which represents the output at that moment, and the cell stage, c(*t*), which retains information from previous time steps h(*t* − 1) and c(*t* − 1). At each time step, c(*t*) is updated by adding or removing information through gates, while the latest input is x(*t*).

An LSTM unit consists of several key components: the input gate, *i*, which processes incoming data by integrating the current input with the previous output and cell state; the forget gate, *f*, which regulates the retention or removal of information from the previous cell state; the output gate, *o*, responsible for determining the final output; and the cell input, *g*, which updates the inputs by incorporating both the current input and the previous output stage. The cell state c(*t*) and the initial state h(*t*), are then computed as follows:(15)$$c\left(t\right)=\;f\left(t\right)⊙c\left(t-1\right)+i(t)⊙g(t)$$(16)$$h\left(t\right)=o\left(t\right)⊙\sigma_{c}(c(t))$$
where ⊙ denotes the Hadamard product, representing element-wise multiplication between two vectors. At time *t*, the states of the LSTM unit components are described by the following equations:(17)$$i\left(t\right)=\;\sigma_{g}\left[W_{i}x\left(t\right)+{RW}_{i}h\left(t-1\right)+b_{i}\right]$$(18)$$f\left(t\right)=\sigma_{g}\left[W_{f}x\left(t\right)+{RW}_{f}h\left(t-1\right)+b_{f}\right]$$(19)$$g\left(t\right)=\sigma_{c}\left[W_{g}x\left(t\right)+{RW}_{g}h\left(t-1\right)+b_{g}\right]$$(20)$$o\left(t\right)=\sigma_{g}\left[W_{o}x\left(t\right)+{RW}_{o}h\left(t-1\right)+b_{o}\right]$$
where $\sigma_{g}$ and $\sigma_{c}$ represent the sigmoid and hyperbolic tangent activation functions, respectively. The trainable parameters of the LSTM unit include biases *b*, input weights *W*, and recurrent weights *RW*, indexed according to its corresponding gate.

### 2.4. XGB Architecture

The Extreme Gradient Boosting (XGB) algorithm was presented by Chen and Guestrin in 2016. It is distinguished by its ability to scale beyond a vast number of examples while using far less resource power on a single machine, and as much as a ten times faster computational time compared with other popular solutions [18]. XGB utilizes gradient boosting to improve a weak model by combining it with many other weak models to create a strong model. For small to medium-sized structured data, decision tree-based algorithms such as XGB are currently considered the most effective [29]. This scalability is utilized by gradient boosting in a decision tree ensemble algorithm. The structure of XGB is shown in Figure 4.

XGB calculates predicted value ${\hat{y}}_{i}$ using a tree ensemble model consisting of *K* number of trees for the $x_{i}\;$input feature, which could contain *n* samples within the *i* observation, and $f_{k}$ as the specific tree structure [20]:(21)$${\hat{y}}_{i}=\sum_{k=1}^{K}f_{k}\left(x_{i}\right)$$The objective function L is then calculated as a sum of loss function l on the total number of observations between predicted ${\hat{y}}_{i}\;$and true $y_{i}$ value, and a regularization term Ω, which penalizes model complexity, thus avoids overfitting:(22)$$\;L=\sum_{i=1}^{I}l\left({\hat{y}}_{i}{,y}_{i}\right)+\sum_{k=1}^{K}{\Omega (f}_{k})$$

## 3. Model Development

The investigated heat exchanger is a conventional industrial shell-and-tube type. According to the standards of the Tubular Exchanger Manufacturers Association (TEMA), the heat exchanger is of AES type, class R. The AES code identifies the design configuration of specific components of the heat exchanger:A—Indicates a front head design: channel with removable cover;E—Indicates a shell design: one-pass shell;S—Indicates a rear head design: floating head with backing device.

Class R refers to rugged service applications typically found in the oil and gas industry. The exchanger contains 556 tubes arranged in a square pitch at 45°, with a length of 4877 mm and an outside diameter of 25.4 mm. The design includes one shell-side pass for the hot fluid and two tube-side passes for the cold fluid. Turbulent flow conditions on the shell side are ensured by 13 baffles spaced at 250 mm intervals. Additional heat exchanger characteristics are shown in Table 1.

Table 1 shows a horizontal orientation of the examined heat exchanger, as well as the mediums flowing inside. The cold medium is the crude oil, reduced after the flash column, and the hot medium is the atmospheric residue. Table 1 also shows that the heat exchanger corresponds to a TEMA AES-type exchanger with one shell pass and two tube passes, indicating the need to apply a correction factor, *F*, to the logarithmic mean temperature difference, in order to account for the deviation from the ideal countercurrent flow. *F* is a correlation of two dimensionless temperature ratios [31]:(23)$$R=\frac{{T_{H,i}-T}_{H,o}}{T_{C,o}-T_{C,i}}$$
and(24)$$S=\frac{T_{C,o}-T_{C,i}}{T_{H,i}-T_{C,i}}$$From ratios for one shell and two tube passes, the *F* is calculated as(25)$$F=\frac{\sqrt{\left(R^{2}-1\right)}\;ln\left[\left(1-S\right)/(1-RS)\right]}{(R-1)ln\left[\frac{2-S\left[R+1-\sqrt{{(R}^{2}+1)}\right]}{2-S\left[R+1+\sqrt{{(R}^{2}+1)}\right]}\right]}$$The temperature correction factor, *F*, for the examined process period is close to 1, with a mean of 0.985. This is expected, since the terminal temperature differences are large.

Process data was obtained from the plant’s history database, covering one year of refinery production. The collected data contained two periods of regular plant production divided by an overhaul period. To ensure data quality and relevance, it is important to analyze and select data representing the complete operating regime of the heat exchanger. Consequently, data from the period immediately following the overhaul (after heat exchanger cleaning) was used for model development. This period captured the full working cycle of the process heat exchanger, from start-up after the overhaul to the next process shutdown. Heat exchangers’ input and output process data were collected with a sample time of 5 min, with a count of 61170 data points. This totals slightly above the 7 full months of heat exchanger work. The dataset included hot flow inlet temperature (*T*_H_,_i_), hot flow outlet temperature (*T*_H_,_o_), cold flow inlet temperature (*T*_C_,_i_), cold flow outlet temperature (*T*_C_,_o_), as well as cold flow mass flow (*ṁ*_C_), and hot flow mass flow (*ṁ*_H_). A statistical analysis of the collected process data over the examined period is shown in Table 2:

From the descriptive statistics for the process variables, it can be seen that the cold stream, flowing through the tubes, exhibits high and stable mass flow with narrow inlet and outlet temperatures, which indicates thermal consistency throughout operation. This could also suggest that the fouling is far more likely to initiate on the hot side where the description shows higher variability in both mass flow and temperature.

Data collection from real plants is challenging because it relies on various measurement devices that are often prone to malfunction and inaccurate readings due to the process nature and deviations from designed process conditions. Therefore, the data were preprocessed to remove and replace the missing values and outliers. Fortunately, the occurrence of missing values was limited and appeared only locally in short segments. Therefore, due to the short duration of these gaps, missing values were imputed using the mean value within a local data window. As the data were sampled every 5 min, the temporal resolution was adequate to ensure that local averaging did not obscure relevant time-dependent variations. To ensure data integrity, trends were visually and statistically examined. Although vertical lines in the trends are visible in Figure 5, these do not represent outliers but are rather real dynamic events within the process lasting over 10 data sample points, equaling more than 50 min of process time. Process conditions are visible via trends of process variables in Figure 5.

The potentially misleading outlier appearance arises from the high density of data points in the plots, which visually masks the natural gradual transitions toward local extrema. This interpretation is further supported by the relatively low standard deviation values reported in Table 2. Figure 5 also shows periods of different crude oil batches, as indicated by the shaded regions in the fourth subplot. These segments reflect discrete operational phases corresponding to changes in crude oil type, labeled A through D, and are particularly visible in the cold-stream flow signal. Each transition results in a subtle shift in the baseline mass flow rate, which may arise from differing fluid properties such as density and viscosity. Importantly, these changes occur without abrupt disturbances, indicating controlled adjustments rather than transient instabilities. This level of process stability further justifies treating the early operational phase—particularly within the first crude type A—as a valid baseline for clean-state modeling, given its consistency across both thermal and flow variables.

Through the examined period, there were four different crude oils over the examined period, which, for confidentiality reasons, were labeled as A, B, C, and D. Crudes A and B were used in their pure form, while C and D were prepared as blends incorporating crude A. All available crude oil properties data associated with fouling and measured in the laboratory were collected during the study period. The crude oil properties measured in the laboratory included density (*ρ*), basic sediments and water (*BSW*), asphaltene content (*Asp*), nitrogen content (*NC*), salt content (*SC*), and water content (*WC*). A statistical analysis of the collected laboratory data is shown in Table 3:

Table 3 provides a statistical overview of laboratory-measured fouling-related properties for each crude oil used during the examined period. Crude B stands out with the highest asphaltene content (49.3% *m*/*m*) and salt content (5.06 mg/kg), both known contributors to fouling formation through precipitation and corrosion mechanisms. It also exhibits the highest water content (7.6% *v*/*v*) and variability (STD 11.2), which may further promote fouling under unstable emulsified conditions. In contrast, Crudes C and D, although denser than A and B, show relatively low and consistent asphaltene and salt values, suggesting better thermal stability in the exchanger. These differences align with the flow trends in Figure 5, where the periods corresponding to Crudes C and D (highlighted in green and orange) exhibit more stable cold-stream behavior, while the segment corresponding to Crude B shows increased variability and a distinct rise in hot-stream flow and inlet temperature. This operational adjustment may have been an intentional response to the high fouling potential of Crude B, as inferred from its extreme asphaltene, salt, and water content values reported in Table 3. Increasing the hot side temperature and flow rate could have been aimed at maintaining heat transfer efficiency by enhancing turbulence and reducing deposition rates on heat transfer surfaces. Such mitigation strategies are consistent with plant-level responses to processing problematic crudes, especially those prone to asphaltene precipitation and salt crystallization. The timing of these changes—occurring just after the Crude B period—further suggests a process adaptation on specific crude properties.

After data preprocessing, the development of the LSTM and XGB models was conducted. The models were developed for the estimation of the outlet temperatures of the cold and hot flows, using data when the heat exchanger operated in a clean state throughout the observed process period. The heat transfer coefficient for the clean and dirty heat exchangers is calculated using Equations (26) and (27):(26)$$U_{\mathrm{fouling}}=\dot{m}\cdot c_{p}(T_{H,i}-T_{H,o})/A\cdot F\frac{\left(T_{H,i}-T_{C,o})-(T_{H,o}-T_{C,i}\right)}{\ln((T_{H,i}-T_{C,o})/T_{H,o}-T_{C,i}))}$$(27)$$U_{\mathrm{clean}}=\dot{m}\cdot c_{p}(T_{H,i}-T_{H,o,\mathrm{model}})/A\cdot F\frac{\left(T_{H,i}-T_{C,o,\mathrm{model}})-(T_{H,o,\mathrm{model}}-T_{C,i}\right)}{\ln((T_{H,i}-T_{C,o,\mathrm{model}})/T_{H,o,\mathrm{model}}-T_{C,i}))}$$$U_{\mathrm{fouling}}\;$is calculated based on the measured outlet temperatures, while *U*_clean_ is calculated based on the model prediction outlet temperatures. The fouling resistance is then calculated using Equation (13) using the clean overall heat transfer coefficient calculated from both LSTM and XGB models’ predicted temperatures as well as the ɛ-NTU method. The decrease in heat exchanger efficiency over the examined period can be measured by the difference between the actual efficiency calculated using Equation (9) and the clean heat exchanger efficiency calculated using the temperatures predicted by the model using Equation (26):(28)$$\varepsilon_{clean}=\frac{\left({T_{H,i}-T}_{H,o,\mathrm{model}}\right)}{\left(T_{H,i}-T_{C,i}\right)}$$

The gradually increasing trend of the fouling resistance and the decreasing trend of the efficiency difference should indicate that the performance of the heat exchangers is decreasing due to the formation of deposits over the studied time.

As previously mentioned, the outputs of the model are predicted using the input data from the initiation phase. The initiation phase is the period when fouling has little or no effect on heat transfer in the heat exchanger, and it begins after the process is overhauled and the heat exchanger is commissioned, while the equipment is still in a clean state. Based on discussions with experienced refinery operators, it is assumed that the period of the first 5000 data points, equivalent to approximately two and a half weeks after cleaning, represents the initiation phase.

The first 5000 data points from the initiation phase were used to develop models for the prediction of outlet temperatures. The subsequent 1000 data points were used in the blind test phase. In this phase, the discrepancy between the model calculated and actual temperatures needed to become apparent to proceed with the method. For the development of the LSTM model, the dataset was divided into training 80% and validation 20%. The search for optimal LSTM model hyperparameters is both time-consuming and labor-intensive; therefore, the KerasTuner [32] was employed. KerasTuner is a scalable and easy-to-use hyperparameter optimization framework that simplifies the hyperparameter search designed for researchers. It allows the easy configuration of search spaces with a define-by-run syntax and includes built-in optimization algorithms. The preliminary results suggested using a single hidden layer of LSTM units, with several parameters selected for detailed examination. The number of historical data in the input layer varied from 2 to 24, representing a historical span of approximately 10 to 120 min. The number of units in the hidden layer was examined from 5 to 250 units with a step of 5. Hidden layer activation function was examined between three commonly used tanh, sigmoid, and relu functions.

The inclusion of a Dropout layer was evaluated to determine its necessity in preventing overfitting, thereby improving the generalization of the model, resulting in a better performance on the validation and test sets. Lastly, two different optimizers, Stochastic Gradient Descent (SGD) and Adam were examined to assess their impact on model performance.

GridSearchCV, a function within the scikit-learn library, was employed to find an optimal set of hyperparameters for the XGB model. GridSearchCV performs a search over a user-defined parameter grid to find the optimal combination of parameters, using cross-validation as the evaluation method. The parameter grid consisted of nine parameters with different values. The n_estimator parameter is used to define the number of trees in the ensemble. Increasing the value means higher performance but increased training time and overfitting potential. The contribution of each tree is scaled by the learning_rate parameter, which with a smaller number requires more trees but makes model more robust. The ratio between overfitting and underfitting is defined by the maximum depth of each tree or the max_dep parameter. To control overfitting by restricting the model from learning specific patterns to individual observations, the value of min_child_weight needs to be increased. Parameters subsample and colsample_bytree serve as reductions in overfitting by introducing randomness into the tree by defining the fraction of samples per each tree and the number of features per each tree, respectively. Lower values increase the randomness, while higher values make the model more deterministic. To manage the splitting of a model and thus its complexity, the gamma parameter is introduced. This parameter defines the minimum loss reduction required for further partitioning of a leaf node. A higher gamma value leads to fewer splits, reducing model complexity. Lastly, two parameters, reg_alpha and reg_lambda, serve as regularization terms to prevent the complexity of the model and overfitting by regulating weights within the model. The LSTM and XGB models’ hyperparameters were systematically examined, and the value grid used in the optimization process is shown in Table 4.

All models were developed in Python 3.12.3 using the following libraries: numpy (1.26.4), pandas (2.3.0), scikit-learn (1.5.0), XGBoost (2.0.3), Keras (3.3.3), and matplotlib (3.9.0). Development and training were performed on a Windows 10 system with an Intel(R) Core(TM) i5-9400F CPU @ 2.90 GHz and 8 GB of RAM, without GPU acceleration. Inference time was measured for both models on the full deployment dataset, which begins immediately after the initiation phase used for development. The LSTM model required approximately 4.81 s (cold stream) and 4.35 s (hot stream), whereas the XGBoost model completed inference in 0.047 s (cold) and 0.031 s (hot). These results confirm that both models, particularly XGBoost, are suitable for near-real-time prediction scenarios in process monitoring, considering the sample time of process variables is every 5 min.

To assess model performance, standard statistical metrics were applied. The mean squared error (MSE) and root mean squared error (RMSE) were used to describe the average size of the prediction errors, with RMSE expressing those errors in the same units as the predicted variable. The mean absolute error (MAE) provided a highly intuitive measure of prediction accuracy, focusing on the average of absolute differences without being overly influenced by extreme values. The coefficient of determination (R^2^) was used to evaluate how well the model captures the overall variability of the target variable. In addition, Pearson’s correlation coefficient was calculated to examine the strength and direction of the linear relationship between the predicted and actual values. In addition to the numerical evaluation, model performance was also examined through visual inspection, including error histograms, scatter plots, and comparisons of predicted and actual values over time. This type of graphical analysis allows for a clearer understanding of local discrepancies, sudden shifts, or patterns that may not be evident from numerical metrics alone. It serves as a useful supplement to quantify how the model responds across different parts of the dataset.

## 4. Results and Discussion

This section presents the results of the developed LSTM and XGB machine learning models, as well as the traditional ɛ-NTU method, applied to predict the outlet temperatures of both hot and cold flows in a refinery heat exchanger. This section includes graphical and numerical comparisons of model performance, an analysis of fouling resistance trends, an evaluation of exchanger effectiveness, and an assessment of the impact of thermal inefficiency on furnace operation.

The search for the optimal combination of hyperparameters for the LSTM models was conducted using KerasTuner, configured to evaluate 50 different trials for each historical time step chosen by the algorithm. This approach resulted in testing more than a thousand different combinations of hyperparameters. The tuner’s objective was to minimize the validation mean squared error (MSE), and the parameter combinations presented in Table 5 yield the best overall score. Similarly, out of several thousand potential combinations, the XGB optimal combination of hyperparameters was found using GridSearchCV from the scikit-learn library. The search was designed to find the lowest mean squared error on the training set. The best hyperparameter configuration is also shown in Table 5.

### 4.1. LSTM Outlet Temperatures Model Results

As previously stated, the development of the hot and cold outlet temperature prediction models was conducted using the first 5000 data points, belonging to the initiation phase.

Achieving the highest possible accuracy in the development dataset was crucial to ensure a robust representation of the relationship between the input and output variables. This approach was taken to guarantee that any systematic deviations observed during deployment over extended periods could be confidently attributed to real process changes rather than errors stemming from model development. Figure 6 and Figure 7 represent the cold and hot flow outlet temperature model development performance based on visual error comparison (a), a histogram of errors (b), and a scatter plot (c). The visual comparison of the cold model (Figure 6) demonstrates that the predicted temperatures closely follow the real temperatures, indicating high predictive accuracy. Slight deviations are noticeable in an isolated segment of the validation set.

This portion of the data exhibits more abrupt changes in the real values, where the model struggles to maintain the same level of accuracy observed in other parts of the dataset. This is also confirmed by a slightly higher mean squared error on the validation set shown in Table 6. The histogram of error shows high levels of accuracy with the errors centered around zero. The narrow distribution of errors shows no significant systematic error or bias, with higher errors confirming the deviations shown in the above visual comparison. The model shows a high correlation between the real and model-predicted temperatures, confirmed by the strong clustering around the diagonal line in the scatter plot. Higher errors are observed farther from the diagonal line, indicating the span of temperatures where the model encounters difficulties, typically corresponding to sudden changes in the real value. Overall, the cold flow model shows great performance, with errors reaching up to ±1.5 K, although the majority of errors (90%) are within ±0.3 K. When compared to the mean temperature range, it is an acceptable error of approximately 0.32% yet confirms the high accuracy and reliability of the model.

The numerical results of the residuals presented in Table 6 clearly show that more than 95% of the prediction errors lie within the ±0.3 K interval, demonstrating the high accuracy and robustness of the LSTM model in predicting cold flow temperature.

Hot flow model performance (Figure 7) shows a highly accurate model with the model prediction trend capturing the trend dynamics of the real temperature values. However, the model encounters minor challenges with certain abrupt changes happening near the end of the development dataset in the validation set. Similarly to the cold flow model, this is reflected in the higher mean squared errors on the validation set (Table 6).

The histogram of errors shows a wider distribution of the errors compared to the cold flow model, indicating less consistency in the predictions of the hot flow model. The broader range of errors suggests that the hot model is less accurate than the cold model, which is confirmed by the comparison of the MSE values of the LSTM cold and hot models (Table 7). However, the values *r* and *R*^2^ are slightly higher than the cold model’s values, suggesting that the hot model has a better ability to follow data trends despite individual errors. The cold model has a lower MSE because its actual values fall within a narrower range, leading to more precise predictions. However, the *R*^2^, which measures how well the model explains the variability of the target variable relative to the mean, is lower compared to the hot model because the variance of the cold data is smaller (2.398) compared to the hot side data (12.818). This greater variance allows the hot model to maintain a higher *R*^2^ as it captures more of the data’s variability. Similarly, the Pearson correlation coefficient is higher, as its predictions align more closely with the overall data trend. Despite the lower accuracy, strong diagonal clustering in the scatter plot confirms the overall model’s high performance. The hot model errors range up to ±4.39 K at their highest, giving an error of 0.89% of the mean temperature range. Nevertheless, the majority of errors (over 90%) are within ±1 K, indicating that while the hot model performs well, its accuracy is slightly lower than that of the cold flow model. These results confirm the acceptability of the hot model’s performance but also highlight its slightly lesser accuracy.

According to Table 8, 90.9% of the prediction errors of the LSTM model for the hot flow lie within ±1 K, indicating slightly reduced accuracy compared to the cold flow case but still reflecting reliable model behavior under more variable operating conditions.

The observed deviations in performance between the training and validation sets are minor, suggesting that the models generalize well and are not overfitted. This conclusion is further supported by the use of early stopping, a widely accepted regularization technique that monitors validation performance during training and halts the process once performance begins to deteriorate [33]. This approach reduces the risk of overfitting by preventing the model from continuing to learn noise from the training data. Additionally, care was taken to ensure that both the training and validation sets were sufficiently large and representative, as unbalanced data splits may lead to misleading conclusions about model generalization. In this study, 20% of the development dataset was allocated for validation, which aligns with standard practices in machine learning and provides a reliable basis for model evaluation and early stopping. With high and similar *r* and *R*^2^ values for both the training and validation sets, it is evident that the models interpreted the data very well, further reinforcing that overfitting is not a concern. The LSTM and XGB models’ accuracy metrics for both hot and cold flows are presented in Table 7.

The first 5000 data points, corresponding to the initiation phase, were used for model development, with 4000 for training and 1000 for validation. The test dataset consists of another 1000 data points immediately following this phase. The objective of this test set was to identify the slow ascending difference between the model-predicted and real data, representing the physical meaning of resistance to heat transfer occurring in the heat exchanger. This subset represents approximately 16.7% of the dataset intended for model development. It was purposefully selected to reflect the earliest signs of fouling development, allowing the model’s predictive behavior to be assessed under changing physical conditions. The focus was on capturing the gradual deviation between the model output and real measurements rather than achieving general statistical representativeness. It is worth noting that the entire development set was used for model training and validation, without setting aside a separate test split from this period. The model maintains high predictive accuracy even at the very beginning of the test set, before gradual deviations start to appear. This further supports the fact that the model is well-fitted to the clean phase of operation. The test set is not used to assess model accuracy in the usual way, but to track when the model starts to diverge, which indicates the beginning of fouling.

Figure 8 and Figure 9 show a comparison of the real and model-predicted outlet temperatures for the cold and hot flows after the initiation phase. 

The cold flow model shows a gradual increase in the difference between the predicted and real temperatures, indicating that heat transfer resistance is occurring between the hot and cold mediums. Figure 9 shows the hot flow outlet temperature comparison and a slightly different behavior of the cold flow.

Here, the model predicted values follow the real ones with certain accuracy yet are slightly lower than those on the development set. This should indicate that the heat transfer resistance on the hot side of the heat exchanger is less pronounced during the observed period. It indicates that fouling on the hot side either has not yet started or has minimal influence on heat transfer.

The models are then deployed on the complete cold and hot deployment datasets of roughly seven months of heat exchanger operation. The cold side results are shown in Figure 10 with the corresponding error analysis in Figure 11, while the hot side results are shown in Figure 12 with the corresponding error analysis in Figure 13. The complete dataset spans the development, test, and remaining dataset, covering the period from start-up to the overhaul of the heat exchangers within one operating cycle. The gradual divergence between real and model-predicted outlet temperature values was observed and confirmed for both cold and hot models.

In Figure 10, the cold flow model captures the overall trend and dynamics of the real temperature data but with a positive bias. The model’s predictions show gradually higher temperature values than the real values, suggesting that the cold flow temperature would be higher if the heat transfer was conducted under clean conditions. This behavior implies that there is a growing resistance to heat transfer from the hot to the cold medium. This error trend is more evident in Figure 11a, where the gradual rise in the error is notable, reaching the highest errors of about 2.5 K, when excluding outliers.

The histogram of errors (Figure 11b) further confirms a systematic positive bias with errors generally spread on the positive side. This consistent bias supports the conclusion that resistance to heat transfer is present and gradually increasing. The hot flow model, as illustrated in Figure 12, similarly captures the trend and dynamics of the real process but with a negative systematic bias.

The model-predicted temperatures are progressively lower than the real values, indicating that the hot flow temperatures would be reduced if heat transfer occurred under clean conditions. This suggests a growing resistance to heat exchange between the hot and cold flows. This indication could be used to calculate the overall heat spent to hold the cold medium at the optimal conditions, which directly impacts the financial picture of the refinery. The error analysis of the hot flow outlet temperature shown in Figure 13 highlights the negative bias, with a gradually increasing error trend in the negative direction.

The maximum error reaches approximately −27 K. This significant deviation indicates that the hot side of the heat exchanger is more severely impacted by deposition or heat transfer resistance. The pronounced impact on the hot side can be attributed to chemical fouling, likely driven by higher temperatures. In the case of the examined heat exchanger, the medium (atmospheric residue) contains components such as heavy hydrocarbons, asphaltenes, and resins, which are prone to fouling. These components deposit more readily at elevated temperatures, exacerbating the resistance to heat transfer. This behavior is well-aligned with the known mechanisms of chemical fouling in high-temperature systems. Elevated temperatures promote the thermal destabilization of heavy hydrocarbon fractions, particularly asphaltenes and resins, leading to their aggregation and subsequent deposition on heat transfer surfaces. These mechanisms are well-documented in the crude oil fouling literature and are considered a primary cause of increased thermal resistance in the hot-stream section of heat exchangers [9]. The lower predicted hot outlet temperatures also provide insights into the energy requirements to maintain optimal conditions for the cold medium. This highlights the financial implications for refinery operations, as additional heat input is required to compensate for the reduced heat transfer efficiency caused by fouling. Overall, the results demonstrate the models’ capability to capture the dynamics of the process and provide valuable insights into the progression of fouling, which directly impacts the operational and economic aspects of heat exchanger performance.

The model-predicted outlet temperatures are further used to calculate the overall heat transfer coefficient as it would be if the heat exchanger remained clean throughout the entire period, using Equation (27), while the real outlet temperature values were used to calculate the real heat transfer coefficient based on Equation (26). The *R_f_* was then calculated from the overall heat transfer coefficients, Equation (13), and is shown in Figure 14. It can be seen that the *R_f_* follows a slowly ascending trend through the observed period, ending with a value of approximately 0.004 W^−1^ m^2^ K.

The overall trendline in Figure 14 represents the growth of *R_f_*, capturing the dynamics of the deposition growth and detachment from the surface, resembling the so-called “saw-tooth” fouling type, which is frequent in industrial heat exchangers. This type of fouling dynamics occurs when parts of already accumulated deposits reach a critical deposit thickness, thus breaking off again, with the contributing factor of periodic variations in process conditions [26].

An analysis of the heat exchanger efficiency trend, as shown in Figure 15, reveals the values significantly impacted by high fluctuations in process dynamics, whether operational conditions, disturbances, or changes in fluid properties.

The efficiency ranges from about 0.8 to 0.3, depending on the process dynamics. Despite these fluctuations, it shows a gradual degradation in real efficiency over time, contrasting with the overall efficiency trend derived from the model-predicted outlet temperatures. This suggests that real efficiency is influenced not only by process dynamics but also by additional factors such as fouling and its potential mechanical implications on the heat exchangers. A slow decrease in the actual efficiency can be noticed as a negative error trend. Over time, a clear efficiency drop of about 20% is observed, confirming the influence of the fouling on the heat exchanger’s performance during the examined period (Figure 16).

### 4.2. XGB Outlet Temperature Model Results

The XGB development model procedure followed the same steps in procedure as the LSTM model development. The accuracy of the cold and hot flow XGB models is shown in Figure 17 and Figure 18, respectively.

The residual distribution presented in Table 9 shows that 95.1% of the prediction errors produced by the XGB model lie within the ±0.3 K range, confirming its high accuracy and suitability for cold flow temperature prediction.

The differences between the XGB and LSTM models are noticeable, with the XGB models being more accurate on both the training and validation sets for cold and hot flow models. This observation is confirmed by the model performance metrics seen in Table 8.

As presented in Table 10, 83.6% of the prediction errors of the XGB model for hot flow lie within the ±1 K interval, indicating slightly lower accuracy compared to the cold flow case, but still acceptable performance for practical applications.

Additionally, the XGB models were developed in less time than the more complex LSTM models. Despite the XGB advantages, limitations began to emerge when the models were deployed across the complete dataset. The downside of the highly accurate XGB models lies in their sensitivity to the training data distribution.

By examining Figure 19 and Figure 20, a divergence in the ability of XGB models to capture the process dynamics can be noticed, particularly in regions with sudden changes.

These issues were not observed during the training and validation phases, as the LSTM models showed better adaptability to such dynamics. Consequently, errors arising from sudden changes in the output variables should not be attributed to model development flaws or overtraining.

The overall accuracy of the XGB models remains satisfactory across the entire dataset, and their performance in capturing general trends is reliable. However, challenges arise in segments where the output variable exhibits ranges not encountered in the development datasets. This highlights the inadequate extrapolation capabilities of XGB models. While XGB models excel in the interpolation of data inside the regions defined by the training set, the decision trees struggle to extrapolate outside the boundaries set by the training, as the thresholds on the tree nodes are defined by the minimum and maximum values of specific features encountered during training [34].

Figure 21 shows the *R*_f_ calculated based on the XGB model outlet temperatures. The results are largely consistent with those derived from what the LSTM models indicate as similar overall trends.

However, local peaks are observed on the parts of the data where the XGB model poorly represented due to its inability to extrapolate. These results show that XGB models are capable of capturing the dynamics of fouling resistance in the examined heat exchanger for a defined period. However, their success in this case is attributed to the fact that local, smaller segments of the data fell outside the training range. This alignment does not guarantee that the model will perform equally well for other heat exchangers, as operational conditions and temperature ranges may vary significantly. If monitoring relies on the real-time evaluation of values outside the initiation phase training range, this method becomes unreliable.

Therefore, while the XGB model can effectively handle interpolation within the training data range, its limited extrapolation capability can make it unsuitable for the presented fouling detection method in cases where significant deviations from the training data are expected.

### 4.3. ɛ-NTU Method Fouling Resistance Results

The ɛ-NTU method was conducted to compare the results derived from the developed models. The average ${\bar{NTU}}_{IP}$, calculated as the average NTU value in the initiation phase, assumed to represent the geometric and hydrodynamic characteristics of the clean phase, was used to calculate the clean overall heat transfer coefficient using the relations in Equation (14). Similarly, the real NTU values were used for the calculation of the real overall heat transfer coefficient. The *R*_f_ was then calculated using Equation (13), similarly to the calculations of model-based approaches. Figure 22 shows the *R*_f_ calculated by the ɛ-NTU method.

The vertical line separates the initiation phase from the rest of the examined period data. The comparison between the model-based approaches and the ɛ-NTU method reveals similarities in the increasing trend of fouling. However, the trend shows a higher level of noise, suggesting that either the method is too sensitive or lacks the ability to account for all relevant information from the data. This random variability undermines the method’s reliability. Based on the nature of the trend, it shows less dynamics compared to the model-based approaches. This could indicate that the method is less capable of capturing the real-time process dynamics as well as more complex model-based approaches, particularly in scenarios involving sudden changes or fluctuations. Since the ɛ-NTU method is relatively straightforward to implement and provides some trend information, it can serve as a useful baseline for identifying general trends in fouling resistance and is advantageous due to its simplicity. However, its inability to capture the complexity and dynamics of fouling deposition, as well as its susceptibility to noise, makes it less reliable than more advanced model-based approaches. For applications requiring detailed insights into fouling dynamics, the model-based methods provide superior performance and accuracy.

Although conducted on different or not entirely comparable processes, the obtained results were compared with similar studies based on machine learning models from the literature. Madhu et al. [14] applied an RF-LSTM architecture for fouling prediction in industrial heat exchangers and reported accuracies of 97–99%, supporting the applicability of time-series neural networks in this field. Similarly, Kang [16] demonstrated that RNNs significantly outperform traditional models in fault detection for the refinery process. Biyanto et al. [35] used a NARX neural network to estimate fouling resistance in a refinery heat exchanger, achieving RMSE values of 1.454 °C and 1.067 °C for the hot and cold sides, respectively. In comparison, the LSTM model developed in this study demonstrated similar performance, achieving an RMSE of approximately 0.8 °C and an R^2^ of 0.99 for the hot flow outlet, while the cold flow prediction exhibited an even lower error with an RMSE of about 0.3 °C and the same R^2^ value. These findings confirm the relevance of LSTM-based models for tracking fouling progression in complex industrial environments.

A comparative evaluation of the LSTM, XGB, and ɛ-NTU methods highlights distinct modeling behaviors. The LSTM model captured gradual increases in fouling resistance along with characteristic short-term oscillations, aligning with expected industrial fouling dynamics involving deposit growth and detachment. The XGB model demonstrated a similar overall trend but exhibited local deviations, particularly in regions with operating conditions not fully represented during training, suggesting reduced robustness in extrapolation scenarios. The ɛ-NTU method also indicated an increasing fouling trend but with higher signal variability and lower responsiveness to process fluctuations. While this method remains computationally simple and useful for general trend observation, its sensitivity to noise and lack of integration of full process context limit its interpretive depth when compared to data-driven approaches.

### 4.4. Impact of Thermal Inefficiency on Furnace Operation

For a clearer interpretation of the calculated fouling resistance, the excess CO_2_ emissions resulting from furnace operation were estimated based on the arbitrarily set 95% furnace efficiency. In the lack of actual values, the fuel gas composition used to calculate excess CO_2_ emissions was assumed to contain 20% hydrogen and an equal mix of methane and ethane, typical for the industrial furnace [36]. The calorific value of the fuel gas was taken as 44.22 MJ/m^3^, with an average emission factor of 1.96 kg of CO_2_/m^3^ [37]. The additional heat required for reheating was calculated based on the difference between the model-predicted outlet temperatures representing an ideal (clean) heat exchanger and the actual outlet temperatures of the cold flow. This excess CO_2_ emission estimation provides an approximation of CO_2_ required solely to compensate for the inefficiency of this specific heat exchanger. However, the total CO_2_ consumption of the furnace is also influenced by other heat exchangers in the preheat train, as well as CO_2_ capture efficiency and overall process variability. Figure 23 demonstrates the thermal inefficiency of the analyzed heat exchanger and its cascading impact on furnace operation.

The mean hot side temperature of the heat exchanger, shown as a blue dotted trend, is primarily influenced by the inlet temperature, but it also reflects heat accumulation due to reduced heat transfer to the cold side. In the early stages, temperature elevation enhances heat transfer performance. However, toward the end of the analyzed period, it is indicative that further temperature increases no longer provide any significant benefit in restoring heat exchanger efficiency, as it cannot overcome fouling resistance. Daily excess CO_2_ consumption, represented by the height of the bars and a color gradient from blue (low consumption) to red (high consumption), follows both trends while providing additional critical insight into when the performance of the heat exchanger becomes unsatisfactory. It can be seen that toward the end of the analyzed period, excess CO_2_ values remain persistently high, suggesting that even with an increase in temperature, the system can no longer compensate for the losses, and the additional CO_2_ consumption remains consistently elevated. This trend signifies a critical stage of heat exchanger degradation, marking a threshold beyond which operational intervention becomes necessary.

The analyzed heat exchanger is a part of a preheat train; therefore, its degradation does not only affect its efficiency but also the performance on subsequent heat exchangers in the network. As this exchanger loses effectiveness, downstream units must compensate for its thermal inefficiencies, leading to a cascading increase in overall energy consumption. Consequently, excess CO_2_, although presented here as a direct result of this specific heat exchanger’s inefficiency, also serves as an indicator of the broader impact on the overall preheating system.

Once the heat exchanger reaches a phase where further temperature increases no longer improve efficiency, the resulting increase in CO_2_ emissions can serve as a crucial metric for evaluating the overall system load. These insights aid decision-making on optimization and preventive maintenance strategies, provided that fuel gas composition and furnace performance are known. In this case study, persistently high excess CO_2_ values start at about 2.5 months before shutdown. The total excess CO_2_ emissions for this period were estimated at roughly 175 tons. Based on the average price of EU Carbon Permits per ton of CO_2_ in 2024, the economic impact was approximately EUR 12,000. Had timely intervention been performed, these environmental and economic costs could have been significantly reduced or prevented.

## 5. Conclusions

In this study, LSTM and XGB models for predicting the outlet temperatures of a heat exchanger in the CDU preheat train were compared to the traditional ɛ-NTU method. Both machine learning models show strong predictive capabilities.

The LSTM models exhibited robust performance in capturing the overall trends and dynamics of the heat exchanger processes. Despite their complexity and longer training times, these models were able to predict outlet temperatures with high accuracy, effectively reflecting the real process dynamics. However, the XGB models encountered limitations with sudden process variations in the validation set. Conversely, the XGB models were easier to train, less computationally intensive, and performed well when interpolating within the bounds of the training data. However, they performed poorly when extrapolating to data ranges not represented during training in more dynamic or variable operational conditions.

This study also highlighted the implications of fouling on heat exchanger efficiency. The gradual increases in error trends and systematic biases observed in the predictions of both the LSTM and XGB models indicate a continuous accumulation of deposits on the heat exchanger surfaces. This fouling is likely attributed to the chemical composition of the fluids, which promotes deposition formation. The resulting decrease in heat transfer efficiency, demonstrated by the discrepancies between predicted and actual effectiveness and the effectiveness errors, highlights the effect of fouling on operational performance.

Additionally, this study examined the impact of fouling on excess CO_2_ emissions, estimating that roughly 175 tons of excess CO_2_ were emitted over the last two months due to the degraded performance of the heat exchanger. Based on the average 2024 EU Carbon Permit price per ton of CO_2_, the economic impact of these emissions is estimated to be approximately EUR 12,000. Had corrective action been taken earlier, these environmental and financial costs could have been avoided. The trend of excess CO_2_ consumption serves as a valuable indicator for assessing system load and decision-making in process optimization and preventive maintenance, provided that fuel gas composition and furnace performance are well-understood.

The comparison of model-based approaches with the ɛ-NTU method revealed significant differences in their ability to capture fouling dynamics. While the ɛ-NTU method provided a simpler and more stable trend of fouling resistance, it lacked the sensitivity and complexity required to accurately reflect the dynamic process behavior captured by the LSTM and XGB models. This suggests that although traditional methods like ɛ-NTU are valuable for their ease of implementation, they may not be as effective as more advanced methods in monitoring fouling in industrial heat exchangers.

This paper suggests that the LSTM and XGB models offer valuable insights into heat exchanger performance and fouling prediction. However, the choice between these models should be guided by the specific needs of the application. LSTM models may be more suitable for scenarios requiring detailed trend analysis and the ability to capture complex dynamic processes. On the other hand, XGB models may be preferred for quicker, less resource-intensive applications within well-defined and stable data ranges.

Despite the promising results, this study has certain limitations related to the nature of real industrial data. Sensor readings, as is often the case in empirical settings, can occasionally be affected by hardware issues, leading to potential noise or bias that is difficult to eliminate entirely. In addition, the operational conditions of the unit vary depending on the type of crude oil processed, which significantly influences fouling behavior. This means that although the developed models performed well on the current dataset, retraining may be necessary when applied to different feedstocks or operating regimes. Still, the applied methodology from data preparation through model evaluation remains robust and transferable across similar cases. Future research should place greater emphasis on model interpretability. While the current study demonstrates strong predictive performance, understanding why the model makes certain predictions remains equally important for practical deployment in industrial environments. Deeper post hoc analyses could help identify which variables most influence the predictions and how these effects change over time. This would not only support model validation but also offer valuable process insights that are often hidden behind purely numerical performance metrics.

## Figures and Tables

**Figure 1 sensors-25-04936-f001:**
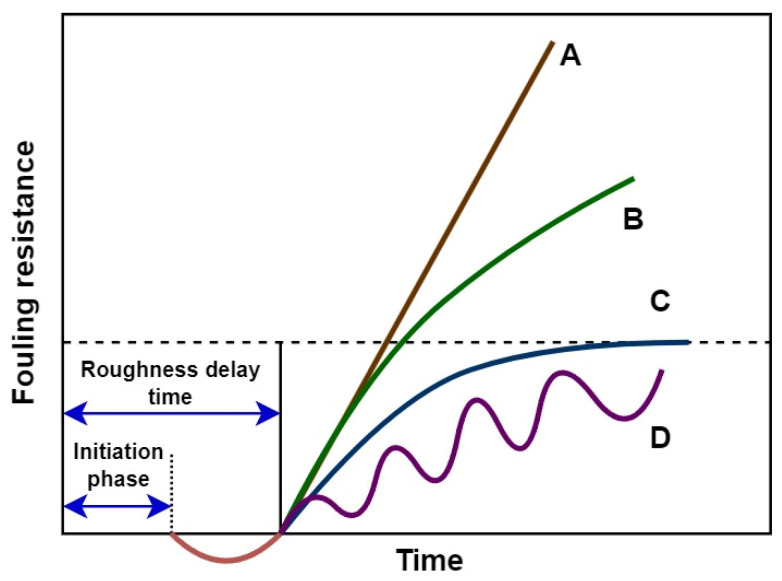
Typical fouling curves [26].

**Figure 2 sensors-25-04936-f002:**
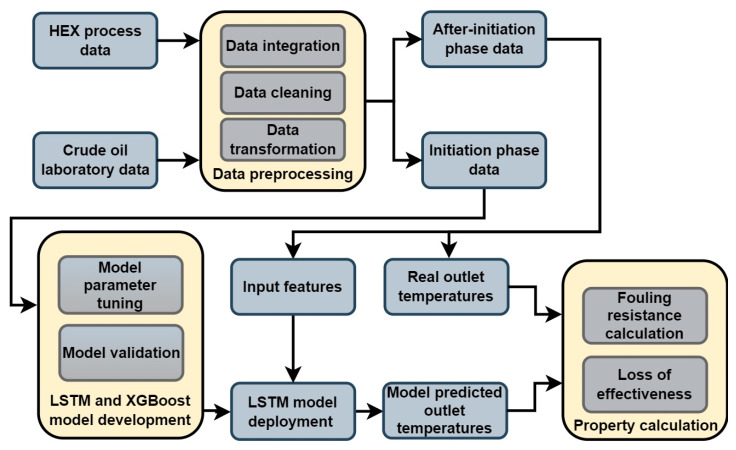
The procedure of LSTM and XGB-based fouling and effectiveness calculation method.

**Figure 3 sensors-25-04936-f003:**
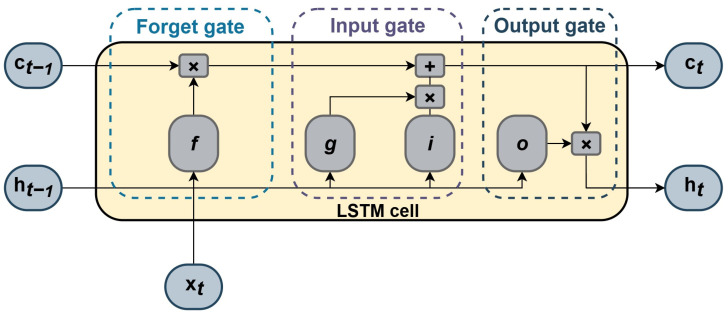
Typical LSTM unit structure [28].

**Figure 4 sensors-25-04936-f004:**
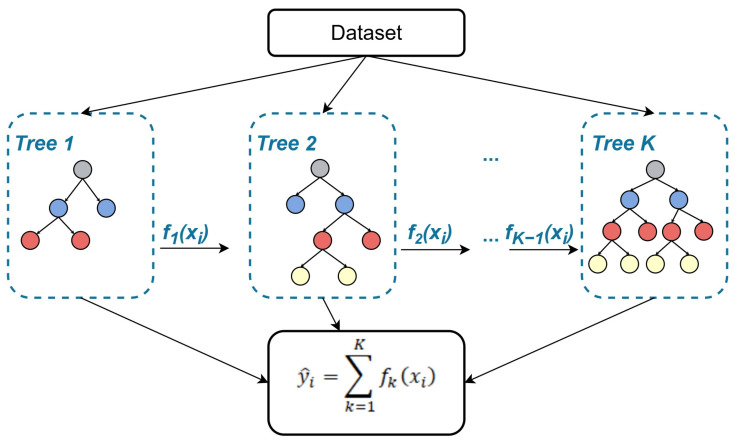
A general architecture of XGB [30].

**Figure 5 sensors-25-04936-f005:**
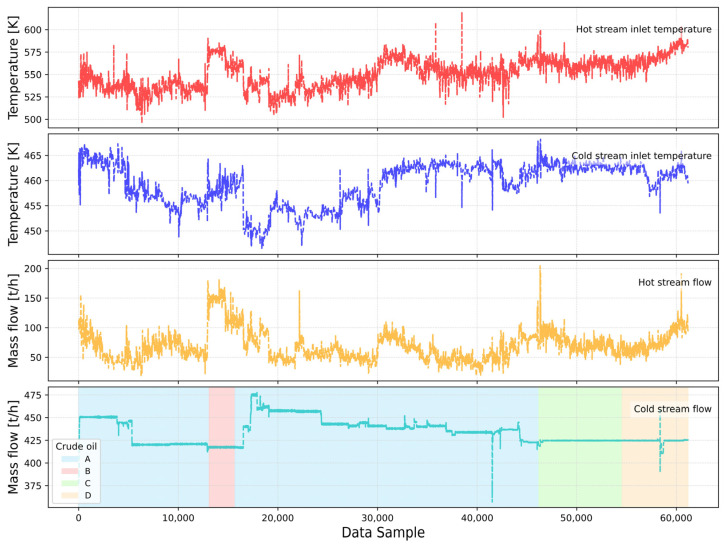
Trends of the inlet process variables.

**Figure 6 sensors-25-04936-f006:**
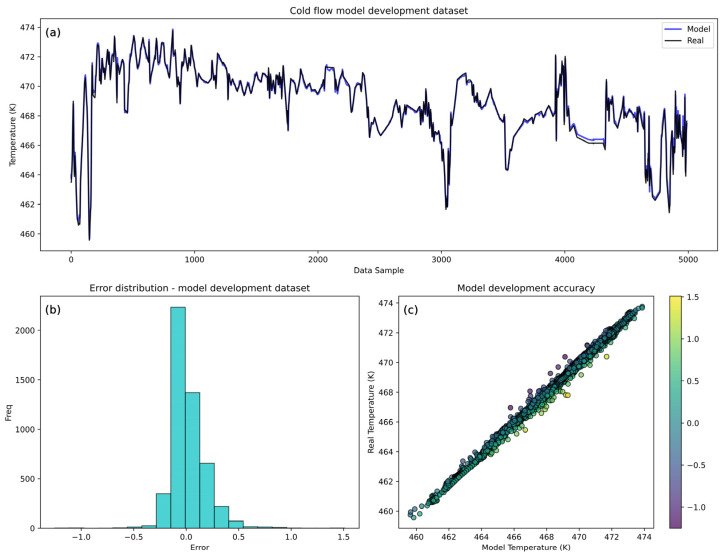
LSTM model training dataset accuracy analysis for cold flow. (**a**) Comparison between real and model-predicted temperatures. (**b**) Cold flow model histogram of errors. (**c**) Scatter plot of real vs. model-developed temperatures.

**Figure 7 sensors-25-04936-f007:**
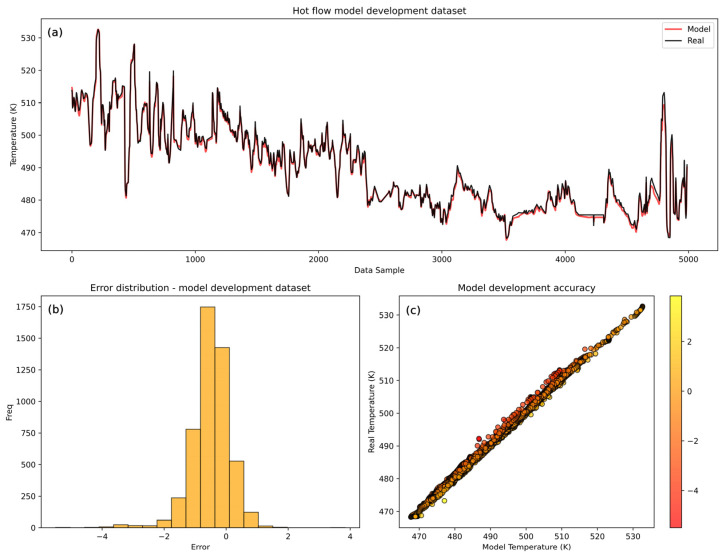
LSTM model training dataset accuracy analysis for hot flow. (**a**) Visual comparison between real and model-predicted temperatures. (**b**) Hot flow model histogram of errors. (**c**) Scatter plot of real vs. model-developed temperatures.

**Figure 8 sensors-25-04936-f008:**
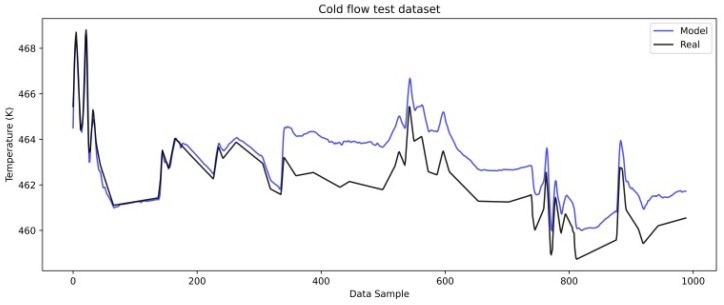
LSTM model test dataset cold flow.

**Figure 9 sensors-25-04936-f009:**
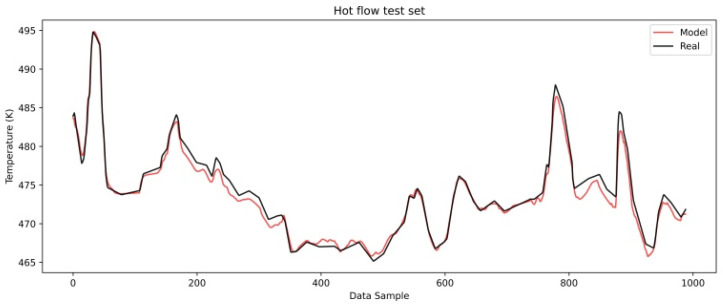
LSTM model test dataset hot flow.

**Figure 10 sensors-25-04936-f010:**
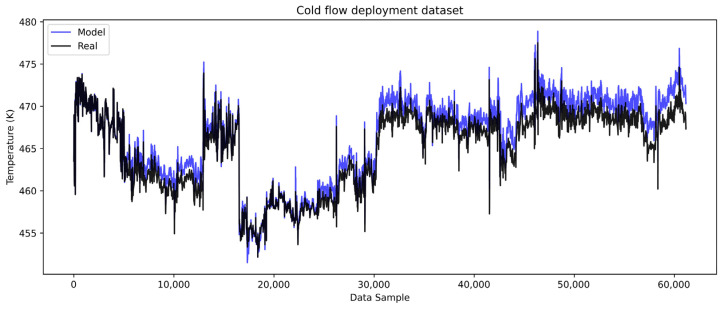
LSTM model complete dataset cold flow.

**Figure 11 sensors-25-04936-f011:**
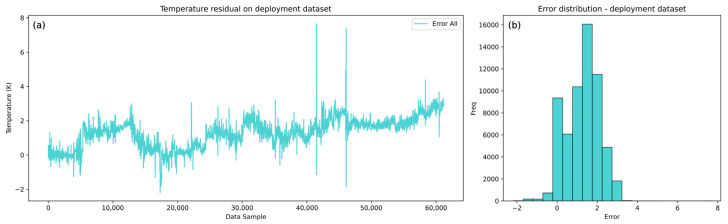
LSTM model complete dataset cold flow outlet temperature error analysis. (**a**) Residual of errors on the deployment dataset. (**b**) Histogram of errors for the deployment dataset.

**Figure 12 sensors-25-04936-f012:**
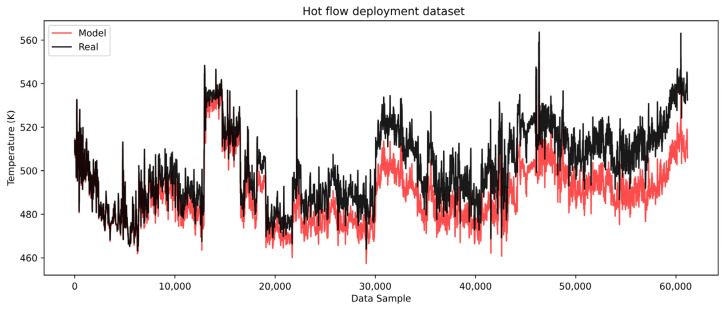
LSTM model complete dataset hot flow.

**Figure 13 sensors-25-04936-f013:**
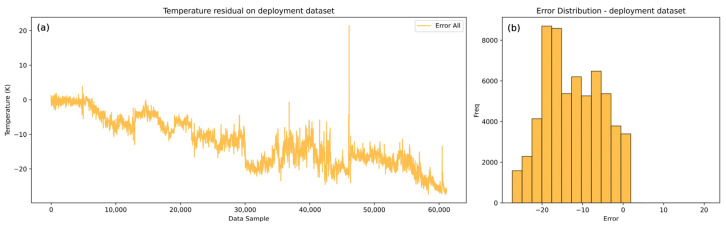
LSTM model complete dataset hot flow outlet temperature error distribution. (**a**) Residual of errors on the deployment dataset. (**b**) Histogram of errors for the deployment dataset.

**Figure 14 sensors-25-04936-f014:**
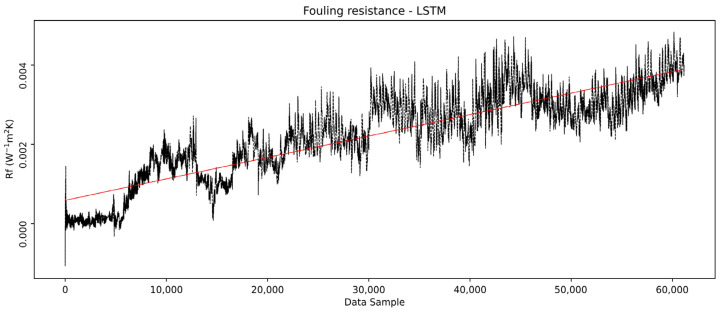
LSTM model calculated the fouling resistance.

**Figure 15 sensors-25-04936-f015:**
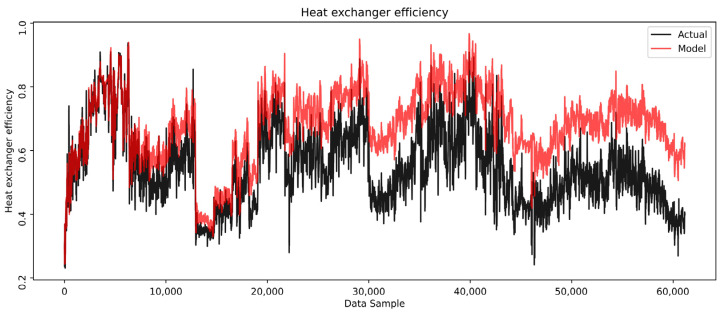
Heat exchanger effectiveness.

**Figure 16 sensors-25-04936-f016:**
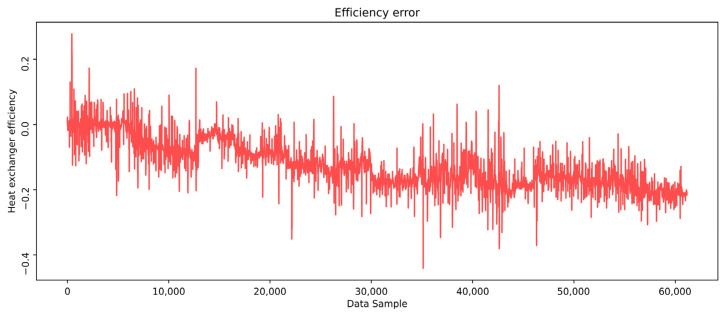
Effectiveness loss.

**Figure 17 sensors-25-04936-f017:**
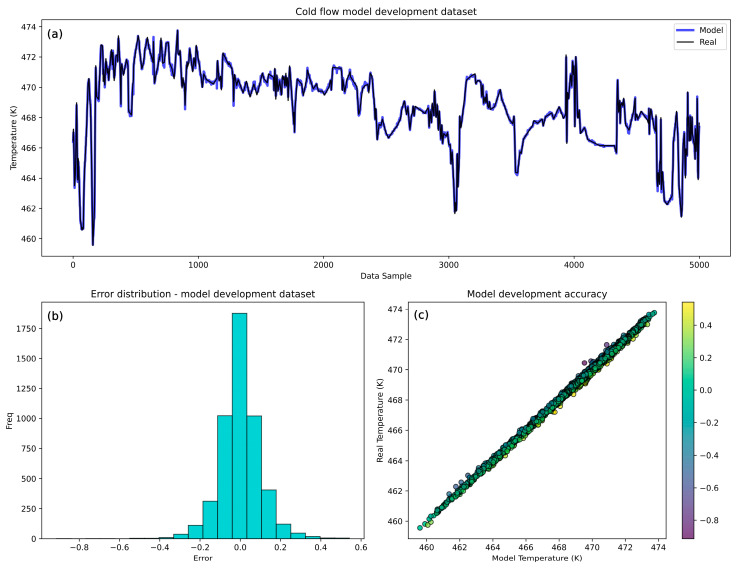
XGB model training dataset accuracy analysis for cold flow. (**a**) Visual comparison between real and model-predicted temperatures. (**b**) Cold flow model histogram of errors. (**c**) Scatter plot of real vs. model-developed temperatures.

**Figure 18 sensors-25-04936-f018:**
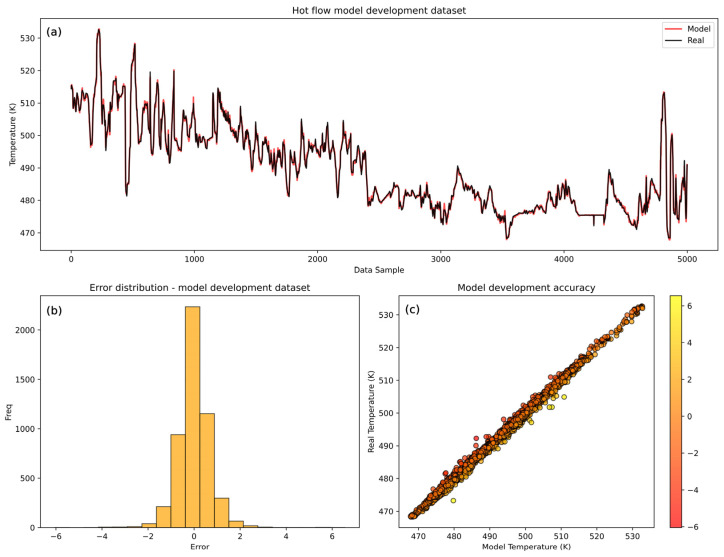
XGB model training dataset accuracy analysis for hot flow. (**a**) Visual comparison between real and model-predicted temperatures. (**b**) Cold flow model histogram of errors. (**c**) Scatter plot of real vs. model-developed temperatures.

**Figure 19 sensors-25-04936-f019:**
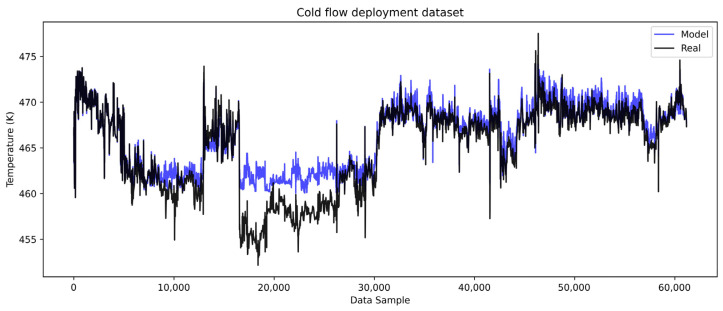
XGB model complete dataset cold flow.

**Figure 20 sensors-25-04936-f020:**
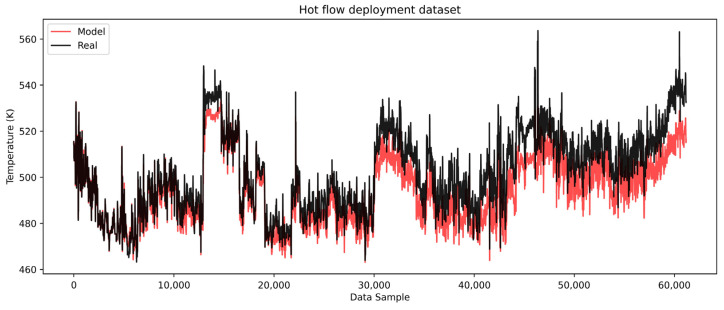
XGB model complete dataset hot flow.

**Figure 21 sensors-25-04936-f021:**
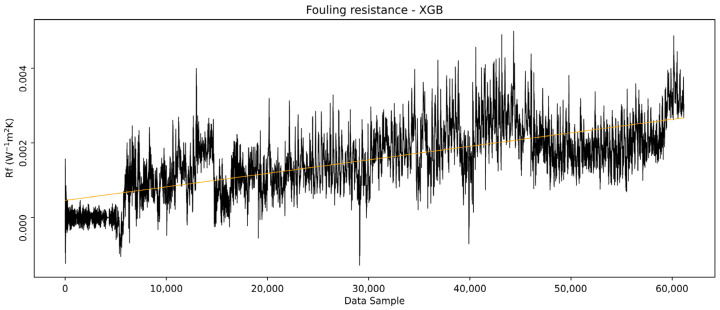
XGB model calculated the fouling resistance.

**Figure 22 sensors-25-04936-f022:**
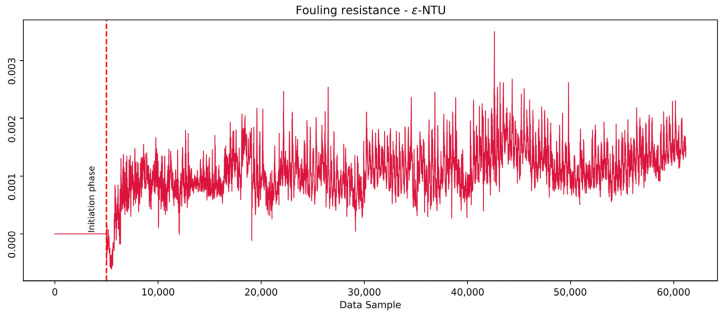
ɛ-NTU model calculated the fouling resistance.

**Figure 23 sensors-25-04936-f023:**
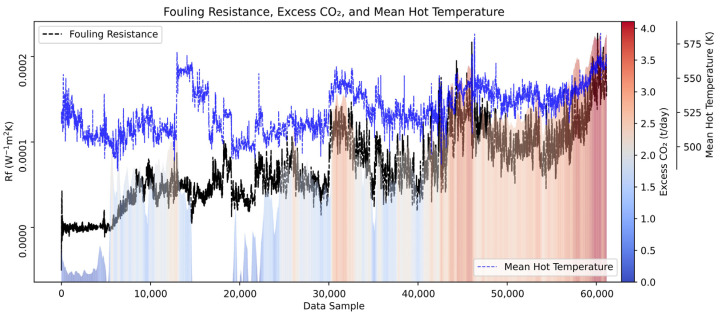
Analysis of thermal inefficiency and its impact on furnace operation due to a fouled heat exchanger.

**Table 1 sensors-25-04936-t001:** Geometric characteristics of investigated heat exchanger.

Parameter	Value	Unit
Type	AES	-
Orientation	Horizontal	-
Medium (hot)	Atmospheric residue	-
Medium (cold)	Reduced crude	-
Area	207	m^2^
Tube length	4877	mm
Number of tubes	556	-
Tube diameter	25.4	mm
Shell diameter	940	mm
Shell passes	1	-
Tube passes	2	-
Shell material	Carbon steel	-
Tube material	Carbon steel	-
Number of baffles	13	-
Baffle spacing	250	mm

**Table 2 sensors-25-04936-t002:** Statistical analysis of the process data over the examined period.

Data	Count	Mean	STD	Min	Max
*ṁ*_H_/kg/s	61,170	68.788	24.313	19.223	204.43
*ṁ*_C_/kg/s	61,170	434.399	13.394	356.943	477.784
*T*_H_,_I_/K	61,170	550.762	16.218	496.635	618.826
*T*_H_,_o_/K	61,170	501.811	17.106	463.210	563.588
*T*_C_,_I_/K	61,170	459.367	4.124	446.420	468.213
*T*_C_,_o_/K	61,170	465.084	4.575	452.168	477.513

**Table 3 sensors-25-04936-t003:** Statistical analysis of the crude oil laboratory data over the examined period.

Crude Oil	A (Mean ± STD)	B (Mean ± STD)	C (Mean ± STD)	D (Mean ± STD)
*ρ*/kgm^−3^	839.2 ± 4.7	845.0 ± 3.8	852.2 ± 1.8	852.1 ± 2.1
*Asp*/% (*m*/*m*)	14.9 ± 16.4	49.3 ± 23.8	5.8 ± 2.2	5.0 ± 0.0
*BSW/*% (*v*/*v*)	4.7 ± 10.4	5.3 ± 2.3	2.5 ± 1.6	4.6 ± 6.0
*WC/*% (*v*/*v*)	3.7 ± 4.4	7.6 ± 11.2	3.6 ± 4.3	3.9 ± 2.6
*NC/*mgkg^−1^	1.24 ± 0.13	1.06 ± 0.05	1.31 ± 0.06	1.20 ± 0.04
*SC/*mgkg^−1^	1.73 ± 1.79	5.06 ± 7.02	0.90 ± 1.25	1.28 ± 0.89

**Table 4 sensors-25-04936-t004:** Examined hyperparameters and the value grid.

**Model**	**Parameter**	**Value**
LSTM	History data	2 to 25 (step = 1)
	Units in hidden layer	5 to 250 (step = 5)
	Activation function	tanh, sigmoid, relu
	Dropout	True, False
	Optimizer	adam, SGD
XGB	n_estimators	100; 200; 300
	learning_rate	0.01; 0.05; 0.1
	max_depth	3; 5; 7
	min_child_weight	1; 3; 5
	Subsample	0.6; 0.8; 1.0
	colsample_bytree	0.6; 0.8; 1.0
	Gamma	0; 0.1; 0.2
	reg_alpha	0; 0.01; 0.1
	reg_lambda	1; 1.5; 2

**Table 5 sensors-25-04936-t005:** Best hyperparameters for LSTM and XGB models.

Model	Parameter	Cold Flow Model	Hot Flow Model
		Value	Value
LSTM	History data	12	12
	Units in hidden layer	135	105
	Activation function	tanh	sigmoid
	Dropout	False	False
	Optimizer	adam	adam
XGB	n_estimators	300	300
	learning_rate	0.1	0.1
	max_depth	3	3
	min_child_weight	1	5
	Subsample	1.0	0.6
	colsample_bytree	1.0	0.6
	Gamma	0	0
	reg_alpha	0	0
	reg_lambda	1	1

**Table 6 sensors-25-04936-t006:** Frequency table of LSTM model results (cold flow).

Error Range (K)	Frequency	Percentage (%)
−1.3 to −1.0	4	0.08
−1.0 to −0.7	4	0.08
−0.7 to −0.3	44	0.88
−0.3 to +0.0	2456	49.23
0.0 to +0.3	2303	46.16
+0.3 to +0.7	151	3.03
+0.7 to +1.0	20	0.40
+1.0 to +1.3	6	0.12
+1.3 to +1.7	1	0.02

**Table 7 sensors-25-04936-t007:** Model accuracy metrics comparison.

Model	LSTM		XGB	
Metric	Cold	Hot	Cold	Hot
Training MSE	0.045	0.443	0.010	0.480
Validation MSE	0.154	0.787	0.009	0.603
Training RMSE	0.212	0.666	0.100	0.693
Validation RMSE	0.392	0.887	0.095	0.777
Training MAE	0.206	0.441	0.081	0.452
Validation MAE	0.408	0.554	0.071	0.418
Training R^2^	0.992	0.998	0.998	0.997
Validation R^2^	0.952	0.985	0.998	0.989
Training *r*	0.998	0.998	0.999	0.999
Validation *r*	0.994	0.997	0.999	0.995

**Table 8 sensors-25-04936-t008:** Frequency table of LSTM model results (hot flow).

Error Range (K)	Frequency	Percentage (%)
−1.3 do −1.0	66	1.32
−1.0 do −0.7	171	3.43
−0.7 do −0.3	390	7.82
−0.3 do +0.0	946	18.96
+0.0 do +0.3	1360	27.26
+0.3 do +0.7	1076	21.57
+0.7 do +1.0	590	11.83
+1.0 do +1.3	200	4.01
+1.3 do +1.7	76	1.52

**Table 9 sensors-25-04936-t009:** Frequency table of XGB model results (cold flow).

Error Range (K)	Frequency	Percentage (%)
−1.3 to −1.0	1	0.02
−1.0 to −0.7	2	0.04
−0.7 to −0.3	32	0.64
−0.3 to +0.0	2471	49.42
+0.0 to +0.3	2453	49.06
+0.3 to +0.7	39	0.78
+0.7 to +1.0	2	0.04
+1.0 to +1.3	0	0.00
+1.3 to +1.7	0	0.00

**Table 10 sensors-25-04936-t010:** Frequency table of XGB model results (hot flow).

Error Range (K)	Frequency	Percentage (%)
−1.3 to −1.0	146	2.92
−1.0 to −0.7	340	6.80
−0.7 to −0.3	671	13.42
−0.3 to +0.0	1241	24.82
+0.0 to +0.3	1240	24.80
+0.3 to +0.7	688	13.76
+0.7 to +1.0	315	6.30
+1.0 to +1.3	138	2.76
+1.3 to +1.7	55	1.10

## Data Availability

The data presented in this study are available on request from the corresponding authors.

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
