# Peer review of "Data-Driven Fouling Detection in Refinery Preheat Train Heat Exchangers Using Neural Networks and Gradient Boosting"

_sensors, 2025, doi:10.3390/s25164936_

Round 1
Reviewer 1 Report
Comments and Suggestions for Authors
-
Line 92: It would be helpful to provide more details on the most common measurement methods, including their advantages and limitations.
-
Section 3 (Model Development):
-
Which software was used for model development?
-
Could you specify the computer configuration (e.g., CPU, RAM, GPU)?
-
Was the execution time of each model measured? A discussion on real-time applicability would be valuable.
-
-
Line 264: Consider adding a figure illustrating the CLT (Central Limit Theorem) in accordance with TEMA to improve clarity.
-
Line 316: Please clarify the rationale behind this methodological choice.
-
Equations 26 and 27:
-
Were experimental comparisons made with other methods or literature data?
-
If so, please include the results; if not, a brief justification would be helpful.
-
-
Line 374: Please explicitly state which model was used in this analysis.
-
Line 457: Providing numerical values would strengthen the discussion.
-
Line 502: Pearson’s coefficient is mentioned but not explained in the Methods section. Please include details on the analysis (e.g., calculated correlations, significance) and where the data can be found.
-
Line 510: Adding references to support or contextualize the results would be beneficial.
-
Line 519: What proportion of the total dataset does the test set (1,000 data points) represent? This would help assess its representativeness.
-
Line 577: Please elaborate on the reasoning behind this finding or choice.
-
General Discussion:
-
The results are promising, but comparisons with other studies (even in different contexts) would enrich the discussion.
-
In the Conclusions, consider addressing:
-
A critical analysis of the study’s limitations.
-
Future research directions (e.g., methodological improvements, potential applications).
-
Reviewer 2 Report
Comments and Suggestions for Authors
In this paper, the authors report three methods for detecting refinery preheat train heat exchangers. There are Long Short-Term Memory (LSTM) neural networks, Extreme Gradient Boosting (XGB), and the É›-NTU method (effectiveness - Number of Transfer Units). In abstract, they claimed that “The LSTM model showed high accuracy in capturing dynamic operational trends, while XGB provided a lightweight alternative with limited extrapolation capability under unfamiliar conditions. Both models outperformed the É›-NTU approach in fouling detection sensitivity.”
The section of the “2. Fouling detection methodology and theory” and “3. Model development is written in detail. Equations 26-28 should be moved to this section.
However, the section on “4. Results and discussion” was challenging to understand. Please rewrite this section.
- What are the numbers of training samples, validation samples, and testing samples? What is the basis for choosing these numbers?
- In “Table 6. Model accuracy metrics comparison”, the inconsistency of the LSTM and XGB was found. Please explain it. The test MSE of XGB of HOT is 5.865. That is unreasonable.
- What are the different meanings of the R2 and r in Training, validation, and testing? The Test R2 of cold in LSTM was 0.293, which is unreasonable.
- Researchers propose many criteria. In this paper, only the MSE and R2 are used. Please explain.
- Please compare three methods, LSTM, XGB, and the É›-NTU method, in the same case.
My suggestion for this paper was to adopt more criteria to evaluate three methods and rewrite the “4. Results and discussion” to make this paper more readable.
Round 2
Reviewer 1 Report
Comments and Suggestions for Authors
All my suggestions have been addressed, and the relevant additions have been appropriately incorporated into the work.
Reviewer 2 Report
Comments and Suggestions for Authors
All problems have been adequately addressed.